# An Intelligent Computer-Aided Scheme for Classifying Multiple Skin Lesions

**Nazia Hameed** [1,*] , **Fozia Hameed** [2], **Antesar Shabut** [3] , **Sehresh Khan** [4], **Silvia Cirstea** [5] **and Alamgir Hossain** [6]

1   School of Computer Science, University of Nottingham, Jubilee Campus, Nottingham NG8 1BB, UK
2   Computer Science Department, King Khalid University, Abha 61421, Saudi Arabia
3   School of Arts and Communication, Leeds Trinity University, Leeds LS18 5HD, UK
4   Department of Computer Science, Shaheed Zulfikar Ali Bhutto Institute of Science and Technology, Karachi 75600, Pakistan
5   Faculty of Science and Engineering, Anglia Ruskin University, Cambridge CB1 1PT, UK
6   School of Computing and Digital Technology, Teesside University, Middlesbrough TS1 3BX, UK
*   Correspondence: nazia.hameed@nottingham.ac.uk

**Abstract:** Skin diseases cases are increasing on a daily basis and are difficult to handle due to the global imbalance between skin disease patients and dermatologists. Skin diseases are among the top 5 leading cause of the worldwide disease burden. To reduce this burden, computer-aided diagnosis systems (CAD) are highly demanded. Single disease classification is the major shortcoming in the existing work. Due to the similar characteristics of skin diseases, classification of multiple skin lesions is very challenging. This research work is an extension of our existing work where a novel classification scheme is proposed for multi-class classification. The proposed classification framework can classify an input skin image into one of the six non-overlapping classes i.e., healthy, acne, eczema, psoriasis, benign and malignant melanoma. The proposed classification framework constitutes four steps, i.e., pre-processing, segmentation, feature extraction and classification. Different image processing and machine learning techniques are used to accomplish each step. 10-fold cross-validation is utilized, and experiments are performed on 1800 images. An accuracy of 94.74% was achieved using Quadratic Support Vector Machine. The proposed classification scheme can help patients in the early classification of skin lesions.

**Keywords:** multi-class skin lesions classification; melanoma classification; acne classification; eczema classification; psoriasis classification; automated classification; skin disease classification

## 1. Introduction

Skin lesions cases are increasing day by day and are a major cause of an increased global disease burden. Skin lesions stand fourth among the major causes of the global disease burden [1]. The after-effects of the skin lesions are severe. The burden of skin lesions is multi-dimensional and includes social, financial and psychological consequences on the patient's life and society [2]. People of all ages suffer from skin diseases, but young and elderly people suffer the most. Unemployment, self-harm, emotional distress, relationship loss, increased alcoholism and suicide are some of the prominent issues found in skin disease patients [3].

A huge difference exists between skin disease patients and the expertise to cope with them. The resources include skilled dermatologists, equipment, medicines and researchers. According to the World Health Organization, people living in rural areas suffer the most because of the lack of resources [4]. Due to this gross imbalance among the skin patients and the expertise, automated expert

systems for early skin lesions classification are required. These classification systems can help in the early diagnosis of skin lesions and help patients living in resource-limited areas.

Fatal vs non-fatal, viral vs bacterial, etc. are some of the categorization of skin lesions. Acne, eczema, psoriasis and melanoma are among the top five most frequently occurring skin diseases [5]. Therefore, this research work investigates the multi-class method aimed at classifying the above-mentioned diseases. This research work is an extension of our already existing work [6]. Acne is a chronic skin lesion and mostly found in adults. It occurs mostly on the face, back and chest [7]. Acne contributes approximately 0.29% in the disability-adjusted life years (DALYs) [8]. The contribution of eczema towards the DALYs is 0.38% [8]. Eczema appears differently in different people. Small patches can be seen in some people suffering from eczema whereas, others may have eczema on full body. Eczema causes the skin to be red, sored, dry and cracked [9]. Malignant melanoma; a kind of skin cancer is a fatal disease and caused by the excessive growth of melanin in melanocytic cells [10]. Malignant melanoma is treatable if detected in the early stages. In 2018, approximately 99,550 new cases of malignant melanoma were diagnosed in the USA and 13,460 of them were incurable; leading to death [11]. Malignant melanoma contributes 0.06% towards the DALYs [8]. Psoriasis is a non-infectious skin problem which causes red patches having white scaly plaques with clear boundaries around them [12]. Contribution of psoriasis in DALYs is 0.19% [8].

Most of the existing work done on skin lesion classification considers a single disease [13–18] and inadequate work is done on multi-class skin lesions classification [19–25]. Due to the similar characteristics of skin diseases, the computational analysis of multi-class classification is very challenging. The core contribution of this research work is a novel intelligent expert classification scheme to classify multiple diseases to provide dermatological care in resource-limited areas. Another contribution of this research work is the bag of features that can be extracted from multiple skin lesions. The proposed classification scheme will be very beneficial for the people living all around the world in classifying skin lesions in their early stages.

The rest of the research article is structured as follows. State of the artwork is reviewed in Section 2. The details of the images and their collecting resources are described in Section 3. The methodology of the proposed classification scheme is presented in Section 4. Results are presented and discussed in Section 5 and conclusion and future work is provided in Section 6.

## 2. Literature Review

Since the 1990s, researchers are working on the automated skin lesions classification [13–19]. The majority of work done in the literature can classify skin tumors [15,26–31] and limited work is done on multi-disease classification [13,14,21–23,32,33]. Within this work, most of the work is performed on the biopsy extracted features [13,21–23,32,33]. Additionally, researchers who have worked on automated extracted features from images just considered single disease classification [19,20,34].

For classifying erythemato-squamous diseases, an automated classification scheme was proposed by Guvenir and his colleague [20] by using three different classifiers. The proposed expert system was trained on the biopsy features and 99.2% classification accuracy was achieved using the voting feature algorithm. Same nature of work was proposed by Ubeyli et al. [21] to classify erythemato-squamous diseases using a combined neural network approach. Their proposed methodology can classify the erythemato-squamous diseases with an accuracy of 97.7%.

Work done by Chang et al. [22] utilize decision tree and artificial neural network(ANN) for diagnosis of same diseases, and an accuracy of 92.62% was attained. For classifying erythemato-squamous lesions on features extracted after a painful method i.e., biopsy; Xie et al. [23], Kumar et al. [24], and Nanni et al. [25], proposed their classification schemes for multi-class skin lesions classification. The classification scheme by Xie et al., achieved an accuracy of 98.61%, whereas the classification accuracy of the other two approaches was 97.22% and 95%, respectively. As stated earlier, the above-mentioned work regarding the erythemato-squamous disease classification was done on the features extracted after a painful procedure, i.e., biopsy [35]. Clinical feature extraction is a

painful, time-consuming and expensive procedure, which requires domain. It is very difficult to extract these features for the people living with limited resources.

To detect malignancy, Erol et al. [36] extracted texture features of the region within the lesion boundary; which was determined by active-contour segmentation. The extracted texture features contain homogeneity, SD, and mean of pixel values. Artificial Neural Network(ANN) and Support Vector Machine (SVM) classifiers were compared and the best performance they achieved was 78% specificity on a dataset consists of 900 images with 173 malignant lesions using ANN. Schnurle et al. [37] provide an automated approach to classify hand eczema. For balancing data, they used the oversampling technique and then extract colour, texture and histogram features from the provided images. For evaluating their approach, SVM was applied to the features extracted from 48 images. An F-score of 58.6% and 43.8% was achieved for the front and back side of hands respectively.

A computer-aided classification system is proposed by Hameed et al. [38] for classification of multiple skin lesions using a hybrid approach in which features are extracted using convolution neural network (CNN) and classification is performed using SVM. As the features are extracted using CNN, hence uninterpretable. Computer-aided classification systems presented by different scholars achieved good accuracy but having the limitation in covering the scope of multiples diseases. Limitations in the current literature indicate the demand for an intelligent classification system that can classify multiple skin lesions with high accuracy.

## 3. Materials

For classifying different skin lesions, dataset plays a vital role. For experiments, an image dataset is collected from different sources. Sources include online medical data repositories, research challenges and researchers working in this domain. The online data repositories include DermIS [26], DermQuest [27], DermNZ [28] and $PH^2$ [29] dataset. "11k hands" publicly available dataset repository is used for healthy images. Some of the images related to eczema and healthy category are collected from researchers [30] working in the field of skin lesions classification. IEEE International Symposium on Biomedical Imaging (ISBI) skin lesion challenge [31] is an international skin lesion classification challenge organized every year since 2016. Some of the images related to benign and malignant class were used from ISBI skin lesions repository. Figure 1 graphically presents the images belonging to different categories. After collecting all the data from different sources, a uniformed dataset has been created for this work.

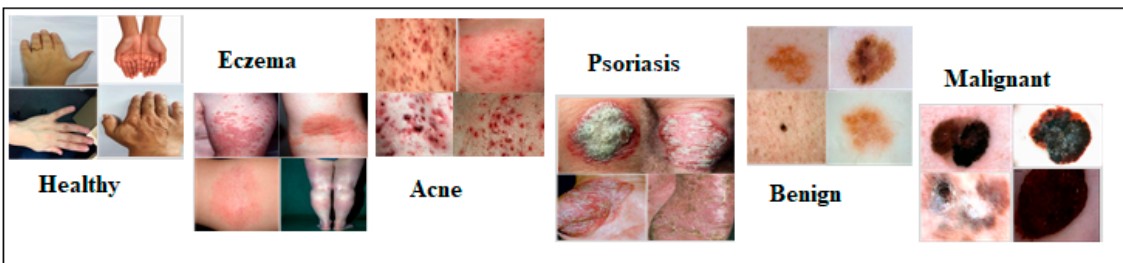

**Figure 1.** Graphical representation of dataset used in the research study.

Data imbalacncing is an important issue that needs to be addressed while training the classification model as the model may incline towards the class having more images [1,32]. Considering this, a stratified sampling technique was used to balance the dataset. Dataset downloaded from the above-mentioned sources is organized based on the disease features and then a random down-sampling technique is applied. Psoriasis category has the minimum number of images (N = 300) so the dataset in other categories is downsampled to make the dataset balanced. After down-sampling, a total of 1800 images of size $227 \times 227 \times 3$ were used to train and test the classification model. Detailed dataset division used in this research work is presented in Table 1.

**Table 1.** Number of images used in healthy, acne, eczema, psoriasis, benign and malignant categories.

| Category | No. of Images |
| --- | --- |
| Healthy | 300 |
| Acne | 300 |
| Eczema | 300 |
| Psoriasis | 300 |
| Benign | 300 |
| Malignant | 300 |
| **Total** | **1800** |

## 4. Method

Pre-processing, segmentation, feature extraction and classification are the key phases of the CAD system for medical image classification [10]. The classification scheme for multi-class skin lesions classification is graphically illustrated in Figure 2, which comprises the phases of preprocessing, segmentation, feature extraction and classification.

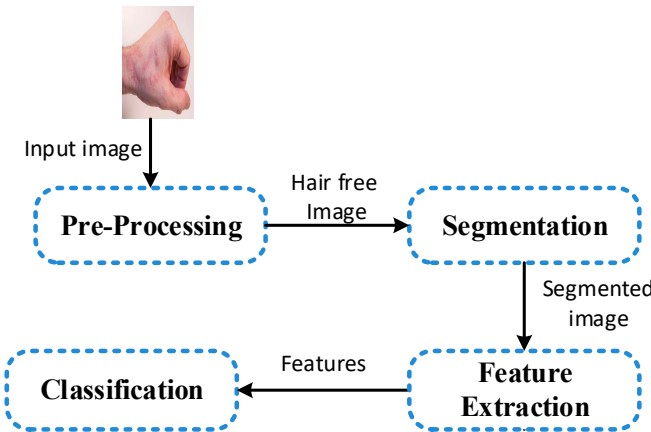

**Figure 2.** Steps involved in the proposed classification framework (1. Preprocessing, 2. Segmentation, 3. Feature Extraction, 4. Classification).

### 4.1. Pre-Processing and Segmentation

Capturing and digitisation is a noisy process considering the facts of angle, lighting, camera resolution and dimensional alignment. Because of the noisy capturing process, pre-processing is the first step of the proposed classification scheme. In this stage, different kinds of noise are removed in the steps of resizing, hair removal and smoothening of the images. The gathered images are of different sizes and contain noise because they are captured using different devices in different environments. The noise present in the images is in the form of hair. As the images are of different size; therefore, for consistency, the images are resized into $227 \times 227 \times 3$. For removing hairs from the images, an already well-known technique titled "Dull Razor" [33] is used. To remove the other noise, a filtering technique is applied and Gaussian filter with $3 \times 3$ filter size is used.

Segmentation of the multi-disease classification is very tough because of their different characteristics and their location on the human body. Malignant melanoma and benign lesions usually have a definite shape and boundary; therefore; shape and geometric features can be easily extracted from them [1]. Diseases like acne, eczema, and psoriasis may cover full body area and have no definite shape, therefore, extraction of geometric and boundary features is very challenging. Due to the above-mentioned problem, in this research, segmentation is performed with respect to human skin. Any non-skin area is discarded from the image and other part is extracted and considered as a region of interest (ROI). ROI is segmented by using the methodology proposed by Phung et al. [34].

The segmentation accuracy achieved is 81.24% as in some cases, the colour of the background and skin matches.

*4.2. Feature Extraction*

Feature extraction for multi-disease classification is a very challenging and difficult task as the different diseases may have similar features. It is also a challenging task due to the diverse nature of the skin lesions, e.g., extraction of shape features is easy from skin cancer images as they have a clear boundary and has a definite size, whereas same features are difficult to extract from acne, eczema, and psoriasis images as they may cover whole body area in the captured image and have no clear shape. In this research work, a bag of features that can be extracted from any skin lesion image is proposed. In the feature extraction step, 35 colour and texture features are extracted from the skin lesion images for multi-class classification.

4.2.1. Colour Features

In multi-disease classification, colour features play a vital role [33,34]. Colour features are one of the important features used to distinguish between different skin diseases. This work explores the RGB colour space, and different features are extracted from it. For this research work, minimum, maximum, mean, mode, standard deviation, skewness, energy, entropy, and kurtosis of red, green, and blue colour spaces are considered. The colour features along with their description and formulae are given in Table 2.

**Table 2.** Different colour features extracted from red, green and blue colour space along with their description and formulae. The colour features include minimum, maximum, mean, mode, standard deviation, skewness, energy, entropy and kurtosis).

| Feature Name | Description | Formula |
|---|---|---|
| Min | Minimum pixel value of R, G and B colour | *Min(colour space)* |
| Max | Maximum pixel value of R, G and B colour | *Max(colour space)* |
| Mean | Measures image overall intensity | $M(\overline{g}) = \sum_r \sum_c \frac{I(r,c)}{M}$ |
| Mode | Gives information about the most occurring value | *Mode(colour space)* |
| Standard Deviation | Presents the spread of the data | $\sigma_g = \sqrt{\sum_{g=0}^{W-1} (g-\overline{g})^2 P(g)}$ |
| Skewness | Measure asymmetry of the probability distribution | $= \frac{1}{\sigma^3} \sum_{g=0}^{W-1} (g-\overline{g})^3 P(g)$ |
| Energy | Gives information about the spread of the pixel values | $= \sum_{g=0}^{W-1} [P(g)]^2$ |
| Entropy | Measure the required amount of information to code the image data | $= -\sum_{g=0}^{w-1} \left[ P(g) \log_2 P(g) \right]$ |
| Kurtosis | Measure of the peakness of the probability distribution of an image | $= \frac{1}{\sigma^4} \sum_{g=0}^{W-1} (g-\overline{g})^4 P(g)$ |

Legends*: $w$ is the number of intensity levels, $g$ is the intensity level, $r$ is the number of rows, $c$ is the number of columns in the image, $\overline{g}$ is the mean, $\sigma_g$ is the standard deviation

4.2.2. Texture Features

In the existing literature, Grey level co-occurrence matrix (GLCM) is mostly used to extract texture features [39]. In this research work, first the GLCM matrix [39] is computed and then contrast, correlation, energy and homogeneity is calculated from it. The extracted GLCM features along with their description and formula are given in Table 3.

**Table 3.** GLCM features with their description and formulae. GLCM features include contrast, correlation, energy and homogeneity.

| Name | Description | Formula |
|---|---|---|
| ContrastGLCM | Measure the local fluctuations of grey levels of neighbor pixels | $\sum\limits_{i,j=0}^{W-1} P_{ij}(i-j)^2$ |
| CorrelationGLCM | Measure the joint probability occurrence of specified pair pixels | $\sum\limits_{i,j=0}^{W-1} P_{ij}\frac{(i-\mu)(j-\mu)}{\sigma^2}$ |
| EnergyGLCM | Measure the sum of squared elements in the GLCM | $-\sum\limits_{g=0}^{w-1}\left[P(g)\log_2 P(g)\right]$ |
| HomogeneityGLCM | Measures the local uniformity | $\sum\limits_{i,j=0}^{W-1}\frac{P_{ij}}{1+(i-j)^2}$ |

Neighborhood grey-tone difference matrix (NGTDM) extracted features are also important and provide the human perception of texture [40]. These features are not fully investigated for the classification of multiple skin diseases. In this research, we have extracted four features from the NGTDM. NGTDM is a column matrix formed by the greyscale image. Let $f(k, l)$ be the grey-tone of any pixel at $(k, l)$ having grey-tone value $i$, the average grey-tone over a neighborhood is calculated using Equation (1).

$$\overline{A}_i = \overline{A}(k,l) = \frac{1}{W-1}\left[\sum_{m=-d}^{d}\sum_{n=-d}^{d} f(k+m, l+n)\right], \ (m,n) \neq (0,0) \tag{1}$$

where $d$ specifies the neighborhood size and $W = (2d+1)^2$ Then the $i$th entry in the NGTDM is calculated using Equation (2).

$$(i) = \begin{cases} \sum|i - \overline{A}_i|, & for \ i \in N_i \ if \ N_i \neq 0 \\ 0 & otherwise \end{cases} \tag{2}$$

where $N_i$ is the set of all pixels having grey tone i. After calculating NGTDM, busyness, complexity, contrast, and strength are extracted. The description along with their formula are given in Table 4.

**Table 4.** Features extracted from the Neighborhood grey-tone difference matrix along with their description and formula.

| Name | Description | Formula |
|---|---|---|
| Busyness | Measure changes in grey levels between neighboring voxels | $= \frac{\sum_{i=1}^{N_g} p(i)s(i)}{\sum_{i=1}^{N_g}\sum_{j=1}^{N_g}|ip(i)-jp(j)|}, \ p(i) \neq 0, p(j) \neq 0$ |
| Complexity | Measure the non-uniformity and rapid changes in grey-levels | $= \frac{1}{N_v}\sum_{i=1}^{N_g}\sum_{j=1}^{N_g}|i-j|\frac{p(i)s(i)+p(j)s(j)}{p(i)+p(j)},$ $p(i) \neq 0, p(j) \neq 0$ |
| Contrast | Measures the changes between voxels and their neighborhood | $= \left(\frac{1}{N_p(1-N_p)}\sum_{i=1}^{N_g}\sum_{j=1}^{N_g} p(i)p(j)(i-j)^2\right)$ $\left(\frac{1}{N_v}\sum_{i=1}^{N_g} s(i)\right)$ |
| Strength | Measure the primitives in an image | $= \frac{\sum_{i=1}^{N_g}\sum_{j=1}^{N_g}[p(i)+p(j)](i-j)^2}{\varepsilon+\sum_{i=1}^{N_g} s(i)}, \ p(i) \neq 0,$ $p(j) \neq 0$ |

All the colour and texture features are stored in the feature vector which is then passed to the classification step for training the classification model.

### 4.3. Classification

Classification is the last phase of the computer-aided classification model. Classification step is the step in which the inferences is made in order to produce a diagnosis about the input image. The classification model is trained on the feature vector using supervised learning. Experiments are performed using different classification models, and the one with the best performance is selected to develop the computer-aided classification application. Different classification models utilized in the classification step are Decision Tree, Support Vector Machine (SVM), K Nearest Neighbor (KNN) and Ensemble methods. For each classifier, different kernels are employed. For decision tree; fine, medium and coarse kernels are used. Linear, quadratic, cubic, fine Gaussian, coarse Gaussian kernels are used for SVM. Kernels for KNN include fine, medium, coarse, cosine, cubic and weighted and for ensemble classifier, boosted trees, bagged trees, subspace discriminant, subspace KNN and RUSBoosted tree kernels are used [41]. The different kernels for each classifier are given in Table 5.

**Table 5.** Different classifiers along with their kernels used in the experiments.

| Classifier | Kernel |
|---|---|
| **Tree** | Fine Tree |
| | Medium Tree |
| | Coarse Tree |
| **Support Vector Machine** | Linear |
| | Quadratic |
| | Cubic |
| | Fine Gaussian |
| | Coarse Gaussian |
| **k-Nearest Neighbors** | Fine |
| | Medium |
| | Coarse |
| | Cosine |
| | Cubic |
| | Weighted |
| **Ensemble** | Boosted Trees |
| | Bagged Trees |
| | Subspace Discriminant |
| | Subspace KNN |
| | RUSBoosted Trees |

The performance of the classifiers is calculated from the confusion matrix. As the proposed CAD system gives multi-class classification, a multi-class confusion matrix is obtained. First, the performance measure of each class is computed, and then the overall performance is calculated. To calculate the performance of the individual class, accuracy, sensitivity, and specificity are used. After calculating the individual class performance, performance of overall classification is computed. Macro averaging [42] is used to calculate the overall performance. The formulae to calculate the overall performance are given in Table 6.

**Table 6.** Performance measures along with their formulae (TP = True Positive, TN = True Negative, FP = False Positive, FN = False Negative).

| Measure | Formula | Description |
|---|---|---|
| *Accuracy* | $\sum_i^l \frac{TP_i + TN_i}{TP_i + TN_i + FP_i + FN_i}$ | Measure the number of correct classifications over the total number of examples evaluated |
| *Sensitivity* | $\frac{\sum_i^l \frac{TP_i}{TP_i + FN_i}}{l}$ | Measure the number of actual positive cases that are correctly identified |
| *Specificity* | $\frac{\sum_i^l \frac{TN_i}{TN_i + FP_i}}{l}$ | Measure the number of actual negative cases that are correctly identified |

Legends:
*i* = Individual class i.e. Healthy, acne, eczema, psoriasis, benign and malignant
*l = Total Number of classes = 6*

## 5. Results and Discussion

The experiments were performed using the gathered dataset and the classification model was trained and tested on 1800 images. K-fold (k = 10) cross-validation technique was used for training and testing the classification model. In k-fold cross-validation, the data is divided into k equal subsets, and the holdout method is repeated k times. Each time, the k$^{th}$ subset is used for the testing and k-1 subsets are used for training, and finally, the average performance across all k trial is calculated. Using 35 colour and texture features, SVM with quadratic kernel performed best among all classifiers. As mentioned above, after performing classification, a multi-class confusion matrix was obtained for each classifier. The confusion matrix for fine tree, quadratic SVM, weighted KNN and bagged trees are provided in the Supplementary Material. The training time required by the SVM with the quadratic kernel was 3.0624 sec whereas the prediction speed was approximately 8400 obs/sec (observations per second). Among decision tree classifiers, fine tree gives the highest accuracy. The average per-class accuracy achieved by fine tree was 88.40%. The sensitivity and specificity obtained by fine tree was 70.24% and 93.04% respectively. The computational time for training the classification model was 3.4608 sec. The maximum number of splits used while using fine tree was 10. As mentioned earlier, among the SVM, Quadratic kernel performed better than others. The accuracy, sensitivity, and specificity achieved by quadratic SVM kernel was 94.74%, 84.23% and 96.85%. The training time for quadratic SVM was 3.0624 sec. For the KNN, weighted KNN performed better with an average per-class accuracy, sensitivity, and specificity of 92.80%, 78.38%, and 95.68% respectively. For weighted KNN, experiments were performed using Euclidean distance and 10 neighbors. The performance of the bagged trees was almost similar to quadratic SVM, and 94.16% accuracy, 82.48% sensitivity, and 96.49% specificity was attained. The results of the fine tree, quadratic SVM, weighted KNN and bagged trees are given in Table 7.

The dispersion boxplot for fine tree, quadratic SVM, weighted KNN and bagged trees is graphically presented in Figure 3 and a comparison of these classifiers is visually presented in Figure 4.

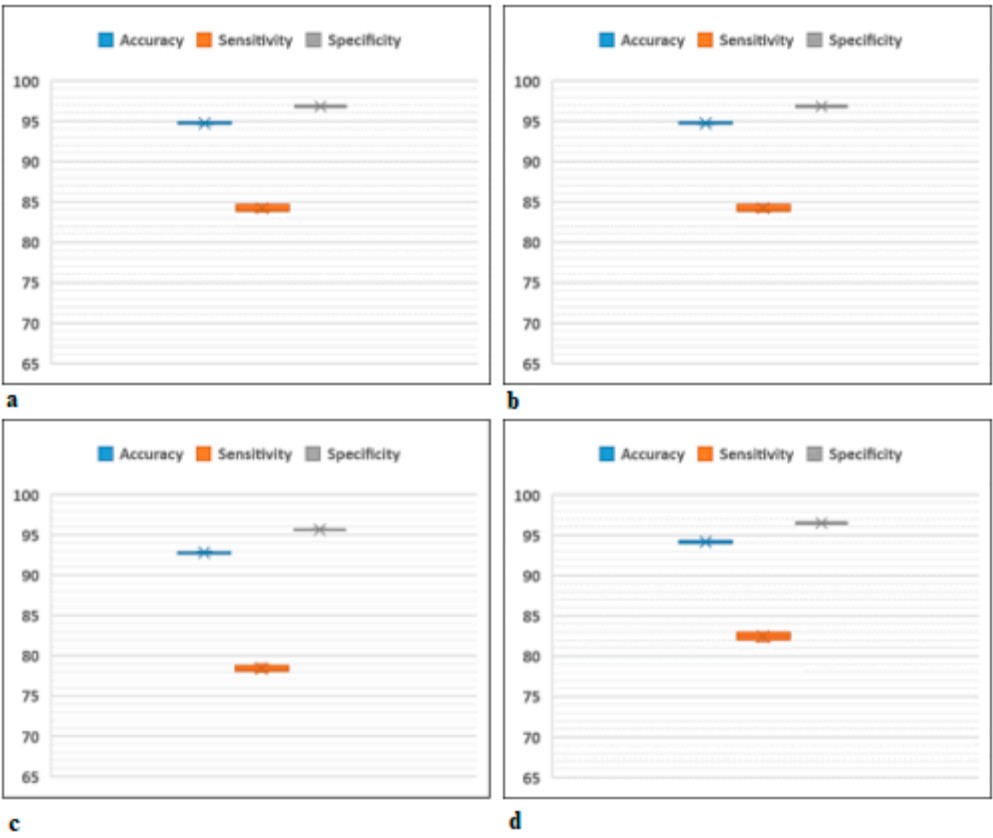

**Figure 3.** Dispersion boxplot for (**a**) Fine Tree (**b**) Quadratic SVM (**c**) Weighted KNN (**d**) Bagged Trees.

**Table 7.** Performance of the different classifiers using the 10-fold cross-validation. Values depict the mean score (Standard deviation). Values in bold show the best accuracy, sensitivity and specificity score. All the score is in %.

| Classifier | Accuracy (SD) | Sensitivity (SD) | Specificity (SD) |
|---|---|---|---|
| Fine Tree | 88.40 (0.27) | 70.24 (0.83) | 93.04 (0.17) |
| Quadratic SVM | 94.74 (0.11) | 84.23 (0.32) | 96.85 (0.06) |
| Weighted KNN | 92.80 (0.11) | 78.38 (0.33) | 95.68 (0.06) |
| Bagged Trees | 94.16 (0.13) | 82.48 (0.39) | 96.49 (0.07) |

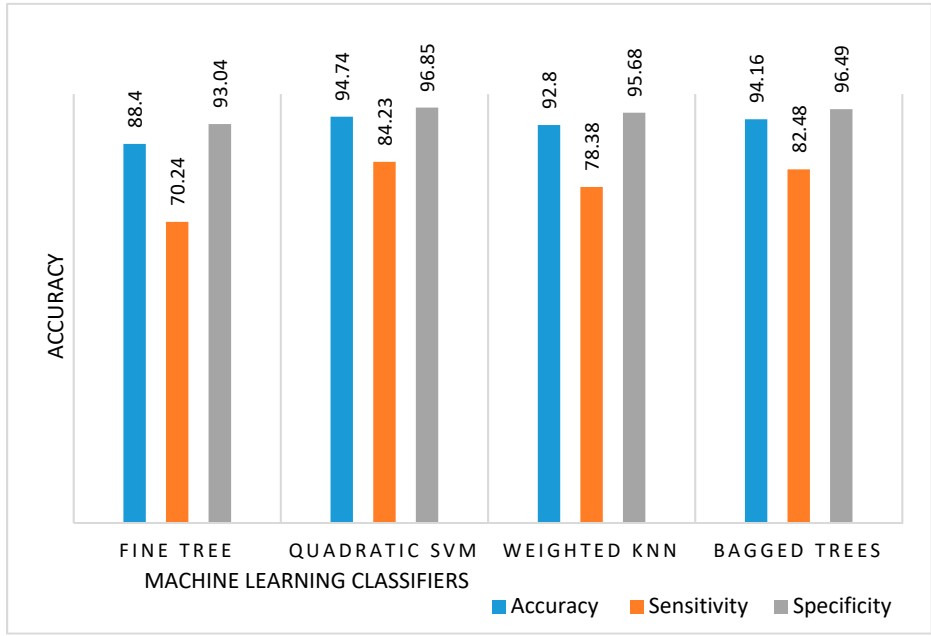

**Figure 4.** Comparison of fine tree, quadratic SVM, weighted KNN and bagged trees.

Based on the performance, the model trained using quadratic SVM is chosen, and the CAD system is developed. Two research works can be compared if they have used the same dataset. The proposed research work is compared with the existing research work and their comparison is illustrated in Table 8.

**Table 8.** Comparison of proposed classification framework with existing research work. All the results are in %.

| Reference | Accuracy | Sensitivity | Specificity |
|---|---|---|---|
| [1] | 83 | NA | NA |
| Proposed Work | 94.74 | 84.23 | 96.85 |

For classifying a new image, an unseen image is sent to the trained model and is classified in a fraction of a second. Currently, the proposed skin lesion classification system can only classify an image into one of the six non-over lapping classes, i.e., healthy, acne, eczema, psoriasis, benign and malignant. If a rarer image arises, it will be classified in one of the provided classes and hence the FPs and FNs will be generated, which can be considered as the limitation of the proposed work. However, it can be overcome by adding more classification diseases. Factors causing difficulties in segmentation and classification are also identified in this work. One of the main hurdles is noise. Noise is present in the form of hairs, black frames, circles, skin lines, etc. Homogenous characteristics of different skin lesions is another reason. Some lesions can have the same colour and texture, which may adversely affect the classification accuracy.

## 6. Conclusions

In the literature, most of the work done on automated skin lesion classification considered only malignant melanoma classification, and the area of multi-class skin lesions classification is neglected. A novel multi-class skin lesions classification framework is proposed in this work for classification of mostly occurred and prominent skin lesions. The proposed framework constitutes four steps; the first step is pre-processing where skin images are pre-processed, and noise is removed from the images. The second step is the segmentation where ROI is extracted from the provided skin lesion image. From the ROI, 35 different features are extracted for the third step, and finally different classifiers are used to train the classification model. Among the different classifiers, SVM with quadratic kernel performed better, with an accuracy of 94.74%. The proposed classification scheme performed very well on the images gathered from different sources. The proposed system can perform very well on new unseen images as it is trained on images collected from different sources.

Segmentation of multi-class skin lesion classification needs more research investigation in order to propose a unified classification scheme that can be applied to different skin lesions images. In this research work, a bag of features was extracted manually, which was time-consuming. Future studies are required for the automated feature extraction which can be easily understandable. The proposed classification scheme is designed for desktop use; more research is required to make this classification compatible with smartphone applications.

**Supplementary Materials:** The following are available online at http://www.mdpi.com/2073-431X/8/3/62/s1.

**Author Contributions:** Conceptualization, N.H., A.S. and A.H.; data curation, N.H. and A.H.; formal analysis, N.H. and S.K.; funding acquisition, A.H.; investigation, N.H., A.S., F.H. and S.K.; methodology, N.H., A.S., F.H. and A.H.; project administration, A.H.; resources, S.C. and A.H.; software, N.H.; supervision, A.S., S.C. and A.H.; validation, N.H. and S.C.; writing—original draft, N.H.; writing—review & editing, N.H., A.S., F.H. and S.K.

**Funding:** This research work was funded by Erasmus Mundus FUSION (Featured eUrope and South asIa mObility Network) project Grant reference number: 2013-32541/001001. Without their financial support it would be not possible for us to carry out this research work.

**Conflicts of Interest:** The authors declare no conflicts of interest.

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
