# Peer review of "An Intelligent Computer-Aided Scheme for Classifying Multiple Skin Lesions"

_computers, doi:10.3390/computers8030062_

Round 1

Reviewer 1 Report

The work is very well presented and written. It is easy to follow and it includes a set of pretty relevant references on the main subject. I recomend its acceptance to the editor after minor typo corrections (as found in lines 63 and 141). My only suggestion is to include a measure of statistic dispersion and highlight best scores in table 7 for better comparison among the classifiers. The figure 3 is unnecesary and referenced erroneously as Fig. 4 in line 255.

Reviewer 2 Report

This is a manuscript about 6 different skin  lesion class image segmentation and classification system.

MAJOR CONCERNS:

1) Given you are using 10 fold cross-validation, I highly recommend you provide performance dispersion boxplot figure plots also together with mean /pm standard deviation table numerical performance values, in results section:

https://en.m.wikipedia.org/wiki/Box_plot

2) You mention confusion matrices but have not been computed in results section. Please provide confusion classification matrices:

https://en.m.wikipedia.org/wiki/Confusion_matrix

Minor comments:

a) Please provide additional detail to figure and table captions whenever possible, thus avoiding potential readers to have to seek through manuscript text.

b) I strongly suggest sharing/including skin image database as supplementary material, at least as 6 different image figures, in order to at least have an idea of the nature of images used.

c) I suggest title modification as follows:  removing "using support vector machine" from title.

Reviewer 3 Report

This paper presents a novel multi-class classification framework to classify healthy, acne, eczema, psoriasis, benign, and malignant melanoma.

My major comments are as follows:

First of all, your result does not match with the title. Your title seems that the proposed CAD only uses an SVM. But in Table 3, you showed the results of Decision Trees, SVM, and KNN. Please be consistent.

Second, the authors mentioned that a random down-sampling was used. Does it mean a stratified sampling? Please specify this.

Third, in Figure 2, you stated that after pre-processing, a noise-free image is obtained. How can you guarantee that the output image is noise-free? Did you check with a signal-to-noise ratio?

Also, please specify the filter size used for Gaussian filters.

Fourth, in Table 5, “Tree” should be “Decision Tree” as it can cause confusion. Often trees mean binary trees.

What do you mean by “Fine Tree, Medium Tree, Coarse Tree, Fine Gaussian, Coarse Gaussian, Fine KNN, Medium KNN, and Coarse KNN?” Does the fine KNN mean 3-KNN? Please explain them in detail.

Fifth, on page 9, what does it mean by the gathered dataset? Does it refer to an image dataset from different sources including images from a public image dataset? Or does it mean your own private dataset? Please explain it in more detail.

Sixth, please compare the performance of the model with those of other research groups in terms of performance metrics you proposed in the manuscript. Also, does it perform consistently across all skin lesions/categories? Please specify these.

Round 2

Reviewer 2 Report

I belive authors have improved manuscript following my suggestions.

Reviewer 3 Report

The manuscript has been significantly improved and can be accepted in present form.